# Food Implications in Central Sensitization Syndromes

**DOI:** 10.3390/jcm9124106

**Published:** 2020-12-19

**Authors:** Elena Aguilar-Aguilar, Helena Marcos-Pasero, Maria P. Ikonomopoulou, Viviana Loria-Kohen

**Affiliations:** 1Nutrition and Clinical Trials Unit, GENYAL Platform IMDEA-Food Institute, CEI UAM + CSIC, 28049 Madrid, Spain; helena.marcos@imdea.org; 2Translational Venomics Group, IMDEA-Food Institute, CEI UAM + CSIC, 28049 Madrid, Spain; maria.ikonomopoulou@imdea.org; 3Institute for Molecular Bioscience, The University of Queensland, St Lucia Campus, QLD 4072, Australia

**Keywords:** central sensitization syndromes, functional somatic syndromes, fibromyalgia, chronic fatigue syndrome, multiple chemical sensitivity, dietary interventions, additives, micronutrient deficiencies, nutritional supplements

## Abstract

Fibromyalgia (FM), chronic fatigue syndrome (CFS) and multiple chemical sensitivity (MCS) are some of the central sensitization syndromes (CSSs). The complexity of their diagnosis, the high interindividual heterogeneity and the existence of multi-syndromic patients requires a multifaceted treatment. The scientific literature is contradictory regarding the role of food in CSS, and evidence on the role of nutrition in MCS is particularly scarce. This review consists in gathering information about the current status of dietary recommendations (i.e., special dietary interventions, the role of additives, presence of micronutrient deficiencies, nutritional supplements and elimination of other nutrients and substances) and discussing the scientific evidence in depth to shed light on appropriate nutritional treatment managements for CSS patients. Current indications show that dietary modifications may vastly improve the patients’ quality of life at a low cost. We suggest personalized treatment, taking into consideration the severity of the disease symptoms, quality of life, coexistence with other diseases, pharmacological treatment, changing clinical characteristics, nutritional status, energy requirements and food tolerances, among others, as the best ways to tailor specific dietary interventions. These approaches will partially overcome the lack of scientific and clinical research on MSC. Patients should also be advised on the serious consequences of following dietary guidelines without a dietitian’s and clinician’s supervision.

## 1. Introduction

Central sensitization syndromes (CSSs) or functional somatic syndromes include different chronic and acquired disorders related to an unknown etiology, such as fibromyalgia (FM), chronic fatigue syndrome (CFS), multiple chemical sensitivity (MCS) and irritable bowel syndrome (IBS).

Although they are considered multifactorial syndromes, hyperreactivity of the central nervous system forms the basis of their origin. Central sensitization induces many functional, molecular and structural modifications. As a consequence, the prospect of establishing a unique definition including all these disorders is remote [1,2]. Several neurobiological aspects have been postulated as the likely routes of pathological models, such as classical conditioning, biochemical disruptions and neurogenic inflammation [2]. These may play a role in a wide spectrum of physiopathological central nervous system responses associated with specific regions of the brain that are connected with emotional and motivational processing of information, like the limbic area (amygdala, hippocampus) and the olfactory and the auditory efferent systems [2,3,4,5]. This situation may explain the development of the physiopathological cascade underpinning CSS, such as autonomic, endocrine and immune disfunction, as well as multisensory perception disruption and some psychological abnormalities of conduct and temper in these patients (anxiety, depression, mood disorders) [3,4].

Consequently, these health conditions share similar physiopathological pathways, even if their symptomatology differs. Central sensitization can generate one or more adverse effects due to a low level of chemical exposure, such as deregulation of the intestinal passage, fatigue or widespread musculoskeletal pain (allodynia or hyperalgesia), among others. In addition, environmental factors and genetic components are involved in the development of CSS. In fact, several single nucleotide polymorphisms related to the presence of FM, CFS and MCS have been described [2,6]. Indeed, it has been hypothesized that specific genetic profiles of impairment in certain essential enzymes are involved in the optimal corporal endobiotic/xenobiotic detoxification (phase I and II), DNA methylation/repair pathways and antioxidant defense/oxidative stress, such as CYP450, GST, NAT2, MTHFR, SOD2, PON1, PON2, NOS2 and NOS3 genes [2]. A case in point, statistically significant differences in the frequencies of MTHFR rs1801133 were observed in a descriptive study composed of 52 MCS patients and 52 healthy volunteers by Loria-Kohen et al. [6]. Similarly, this single nucleotide polymorphism was related to MCS in another study by Micarelli et al. in MCS patients that also found an association with some functional polymorphisms in genes affecting enzyme function, such as MTHFR rs1801131, SOD2 rs4880, NOS3 rs1799983, PON1 rs705379 and rs 662 [2].

The medical diagnosis of CSS is complex and lengthy because the diagnostic criteria are not standardized and because of the high interindividual heterogeneity of their clinical features that depend on stimulation (chemical substances, lights, pressure, sounds, temperature, etc.), the origin of the affected tissue (muscle, skin, the mucosal surface of the digestive tract, etc.) or psychobiological manifestations [1,7]. Multiple CSS is commonly found among patients. Both FM and CFS are characterized by chronic widespread pain (CWP). Their comorbidities could be answered in the common mechanisms of cellular oxidation and are usually investigated together [8,9]. Hence, multi-syndromic patients pose an additional complication for the clinicians [7].

Nowadays, all these difficulties (lack of harmonization of diagnostic criteria, individual differences in reasons for the stimulation and clinical manifestations, multi-syndromic conditions and comorbidity) imply a complexity when establishing the worldwide prevalence of CSS. Interestingly, specialists consider females to be more prone to CSS than males [8,10,11]. The worldwide prevalence of FM is estimated to be 0.2 to 6.6%, 0.8 to 3.3% for CFS [12] and 0.1 to 5% for MCS [13] and continuously rising [14].

Currently, a standard treatment for CSS does not exist [8,13]. Pharmacological and nonpharmacological managements, including physical activity, are considered [15]. The use of dietary components with potential functionality on inflammation (i.e., flavonoids, vitamin D or n-3 fatty acids) and oxidative stress (i.e., magnesium, selenium, zinc, vitamin A, C and E or coenzyme CoQ10) seem promising in the treatment of these patients due to the alterations in oxidative stress and neuroinflammation that they may suffer. However, the scientific literature is contradictory regarding dietary and nutritional management [6,8].

As a result of this, dietitians and clinicians are not able to guide patients correctly. Likewise, many patients seek alternative therapies and diets, which instead of being beneficial, cause a decline in their health and eventually the quality of life (Figure 1).

Altogether, this prompted us to gather information about dietary recommendations’ current statuses and discuss the scientific evidence in detail to shed light on nutritional management for CSS patients.

## 2. Methods

Articles were selected using the online search MEDLINE (via PubMed) database according to Medical Subject Headings (MeSH) terms up to March 2020. Clinical human articles associated with central sensitization disorders, functional somatic syndromes, fibromyalgia, chronic fatigue syndrome and multiple chemical sensitivity diseases were included without distinguishing sex or age. Meta-analyses, systematic reviews and reviews were included (Figure A1), except for CSM, in which all articles (case reports, classical articles, clinical studies, clinical trials, comparative studies, controlled clinical trials, evaluation studies, journal articles, meta-analyses, multicenter studies, observational studies, randomized controlled trials, reviews and systematic reviews) were considered in this review (Figure A2). Publications on irritable bowel syndrome were excluded since this condition is already widely studied.

The keywords used were “((central sensitization disorders OR functional somatic syndromes OR fibromyalgia OR chronic fatigue syndrome OR multiple chemical sensitivity NOT irritable bowel syndrome)) AND (food OR diet OR energy intake OR dietary macronutrients OR dietary carbohydrates OR dietary protein OR dietary fats OR essential fatty acids OR dietary micronutrients OR dietary vitamins OR dietary minerals OR celiac disease OR gluten OR FODMAP OR food hypersensitivity OR food allergy OR food intolerances OR food allergens)”.

Given the limited available scientific literature about MCS, the keywords were further expanded to the following “((multiple chemical sensitivity OR multiple chemical sensitivities OR idiopathic environmental intolerances OR MCS OR idiopathic environmental intolerance)) AND (food OR diet OR energy intake OR dietary macronutrients OR dietary carbohydrates OR dietary protein OR dietary fats OR essential fatty acids OR dietary micronutrients OR dietary vitamins OR dietary minerals OR celiac disease OR gluten OR FODMAP OR food hypersensitivity OR food allergy OR food intolerance OR food allergens)”.

We manually revised and included Spanish and English papers. To select articles to include, the abstracts were first read carefully. Subsequently, the full text of those seemed to be considering the nutritional approach to CSS was accessed in order to verify its suitability. Duplicate references, those without full text access and those involving pharmacological, complementary or alternative treatments, such as mind–body therapies, exercise and alternative medicine, were discarded.

## 3. Dietary Components in These Syndromes

The most relevant diets and micronutrients studied in the literature are summarized in Table 1 and Table 2, respectively.

### 3.1. Special Dietary Interventions

The consumption of specific beverages, food and dietary substances, including food additives, alcoholic beverages, caffeine, casein, corn, eggs, gluten, lactose, nuts, shellfish, solanaceous plant species, soy, yeast and arachidonic acid has been reported to trigger pain in CSS patients [12,14,15,20].

The exclusion or inclusion of specific ingredients or foods and the managing of special diets different from the omnivore pattern are some of the most used CSS patient dietary interventions [15,24]. Some of these dietary interventions could become monotonous and inadequate from a nutritional perspective [13]. Hence, these practices could lead people to malnutrition, either by under- or over-nutrition, as reported in several studies carried out in MCS patients [6].

Hypocaloric diets

Obesity exacerbates FM symptoms, causing pain sensitivity and affecting the quality of life [8]. However, it is unclear if the excess weight is indeed linked with FM [17] and whether hypocaloric diets are beneficial for FM patients [11]. A weight-loss intervention aiming to assess the effect of a low-calorie diet (800 kcal/day) for approximately 12 weeks in 123 obese adult patients improved depressive symptoms, fatigue and the spatial distribution of pain [11]. However, in CFS observational studies, excess weight is not significantly related to the severity of fatigue or functional impairment [24].

Elimination diets

The literature has reported a high rate of food allergies and intolerance in FM, CFS and MCS patients, although their diagnosis has not always been accredited by scientific bodies [8,12,13].

Elimination diets consist of a rotation sequence of food components to identify which cause the exacerbation of symptoms and omit them. One by one, food components must be introduced into the diet every 4-5 days. There is a limited number of studies that have addressed the real effect of this practice in CSS. In 1992, two American clinicians stated that avoiding foods causing aggravating symptoms and controlling the environment might be sufficient to achieve normal immune system function in MCS patients. They proposed to eliminate common foods in traditional regional cuisines. For instance, dairy products, corn, processed foods, refined sugars, sucrose and wheat in American populations or rice and soy in Asian populations [18].

The recommendation in MCS patients to avoid foods through elimination diets should be adequately designed to provide variety and a correct nutritional profile [19]. A recent descriptive cross-sectional trial of 52 patients diagnosed with MCS (61.5% with the cumulative presence of FM and CFS) reported that over half of them followed exclusion diets. This dietary intervention could imply eliminating the consumption of certain foods and, subsequently, a higher risk of macronutrient imbalances and micronutrient deficiencies. Hence, the authors recommended a suitable replacement with better-tolerated foods belonging to the same food group regarding the nutritional composition [13].

A non-randomized and non-blinded study assessed 51 subjects with FM to address an exclusion diet program’s effectiveness on remission symptoms. Eleven individuals were assigned to a control group and 40 to a treatment group. While control subjects maintained their lifestyles, food sensitivities by a lymphocyte response assay were identified in the treatment group. Participants substituted food components with multiple dietary supplements. There was an elevated dropout rate of participants, but patients who completed the 6-month intervention showed improvement in signs of depression, fatigue, pain and stiffness. However, the reported results were inconclusive [17].

Avoidance of food additives

Food additives, including food coloring, monosodium glutamate (MSG) and aspartate, are the primary triggers in FM and MCS [8,14,15]. The hypothesis about the development of CSS states that there is a gradual loss of tolerance induced by frequent exposure to and bioaccumulation of certain toxic substances. Therefore, a multitude of physical or neuropsychological manifestations can be developed. Due to this acquired sensitivity, the deleterious symptoms could be managed if the environmental factors that provoke it, such as certain food additives, are avoided indefinitely or temporarily [14]. Some authors point to MSG and aspartame as excitatory neurotransmitters and, consequently, central nervous system stimuli [14,17]. This fact could be due to the combination of cerebral vasodilation and sensitization of nociceptors [20], which raises the possibility that if suppressed, it could reduce the perception of pain in CSS patients [14,17].

Although it is proven that the intramuscular injection of glutamate into muscles entails muscle pain, the effect of its consumption on the pathogenesis and preservation of musculoskeletal pain remains unknown [8]. MSG is a common flavor enhancer in processed foods. Its overconsumption had been linked with a large increase in serum glutamate concentrations and the development of adverse effects related to pain, including in healthy populations. One review stated that the ingestion of 150 mg/kg per day of MSG over 5-day period, which is equivalent to the average MSG consumption in the Western diet (170–250 mg/kg/day), is associated with headaches and muscle tenderness in a healthy population. This study concluded that low-MSG diets may decrease muscle pain symptoms in FM, although rigorous investigation is still limited and well-designed clinical trials should be conducted [20].

Aspartame is used as an artificial sweetener in some food products and its regular daily consumption (5 mg/kg) is safe in FM patients. However, when this dose is exceeded, aspartic acid, which is one of its metabolized components, has the capacity to aggravate headache [20].

A case report in a 5-year-old girl, who developed immediate severe reactions after the consumption of candies containing tartrazine, implied that azo dyes could trigger MCS. Of note, the mother suffered from MCS, so the authors suggested the likely existence of a mother–child genetic key factor [28]. However, there is a limited number of pediatric cases to verify the conclusions of this study in respect to the effects of tartrazine in MCS children [21].

Well-balanced and varied diets

A current systematic review concluded that a balanced and varied diet is a better recommendation than general long-term elimination diets in CFS patients [12]. Indeed, a review performed by clinicians underlines the importance of following a well-balanced diet and avoiding alcohol, caffeine, deep-fried foods and sugars, as well as eating small meals and snacks [16].

Similarly, Bjørklund et al. highlighted the beneficial effect on FM symptoms of following a healthy and vegetable-rich diet [9].

A longitudinal 5-year follow-up study containing 532 cases with chemical intolerance (CI), represented by intolerance to odorous and pungent chemicals and 1260 controls, based on a self-reported questionnaire survey, revealed that maintaining a regular lifestyle with appropriate physical activity, diet and sleep was sufficient for improving disorders of the nervous system (i.e., depressed mood, irritability, fatigue, anxiety or somatic symptoms). However, although MCS is often referred to as severe CI, precise disease definition criteria are not available, and these results may not be extrapolated to these patients [22].

Oligoantigenic diets

Another diet considered in CSS patients is the oligoantigenic, which consists in disregarding processed foods and replacing them with the paleolithic equivalents. However, these diets lack scientific evidence, and should not be recommended [8].

Vegan and vegetarian diets

Many studies have emphasized the positive repercussion of following a healthy diet enriched in vegetables on alleviating symptoms of FM [9]. The effects of vegan and vegetarian diets have been praised due to their antioxidant power. The reduction of oxidative stress may diminish pro-inflammatory cytokine levels, and improve FM and CFS symptoms [9,24].

The living foods diet (LFD) is a raw vegan diet that contains a high quantity of antioxidants [8], including berries, cereal, fruits, germinated seeds, nuts, roots, sprouts and vegetables. In addition, it is free of alcohol, coffee, table salt and tea [15]. The LFD may improve joint stiffness and pain in FM patients due to higher levels of polyphenolic compounds (i.e., kaemferol, myricetin and quercetin, and vitamins such as A (α and β-carotenes), C and E) than their counterpart controls [8].

Some of the LFD ingredients are fermented, so the diet could be considered rich in lactobacteria. The impact of dietary modifications can alter the gut microflora, so it is essential to investigate these diseases’ implications due to its significant role in the immunological defense, such as the bidirectional communication through the gut–brain axis between the enteric and central nervous systems. In a 3-month non-randomized, controlled study, the LFD showed beneficial effects on symptoms, such as morning stiffness, pain scores and sleep quality, as well as in health assessment questionnaire scores in FM patients. However, it was also recorded that FM subjects had lost weight. This fact may have influenced the outcomes because of the prevalence of locomotor problems and the improvement that losing weight can have in previously overweight or obese individuals. Nevertheless, randomized clinical trials (RCTs) are still required [8,17].

In a 7-month follow-up interventional study which followed a raw vegetarian diet based on carrot juice, cereal, fruits, nuts, salads, seeds and tubers in 30 FM patients but without the appropriate control group, the authors concluded that this diet could improve all scales and criteria of a quality of life survey (general health, mental health, physical functioning, emotional role, physical role, social functioning and vitality), except body pain or the impact of fibromyalgia according to Fibromyalgia Impact Questionnaire measurements of their lives [8,17].

Some studies have compared the effect of vegan or vegetarian diets with pharmacological treatments [8,17,23]. Regarding some reviews, a 6-week RCT with two arms evaluated the effects of a vegetarian diet exclusively in comparison to the administration of a tricyclic antidepressant (10–25 mg/day, titered up to 100 mg/day during the study) in 78 FM patients. A decrease in pain scores was observed in both groups, but with a lower trend for the diet group [8,17,23]. Other variables, like insomnia, fatigue and non-restorative sleep, were improved in the treatment group. In fact, all the patients in the vegetarian diet group chose to continue with their pharmacological treatment when the study was concluded. The authors determined that further studies were necessary to prove the effectiveness of this diet, as an exclusive treatment for FM [8,17,23].

Low-FODMAP diets

The concurrence of FM and IBS is estimated to be up to 81% [15]. Several recent meta-analyses concluded that dietary modifications through the short-term restriction of foods rich in fermentable, oligo-, di- and monosaccharides and polyols (FODMAP) show benefits in IBS patients [36,37]. The overlap of both diseases may be linked with the perception that lactose is a frequent trigger in FM patients [15]. A low-FODMAP diet has been examined in an 8-week study of 38 patients with FM but without a control group. A high percentage of patients reported an improvement in abdominal and somatic pain [11].

Gluten-free diets

Sometimes, certain gastrointestinal symptoms in FM patients are related to gluten-related disorders distinct from celiac disease, such as non-celiac gluten sensitivity (NCGS), an underlying treatable cause of FM [38]. Indeed, a prospective study of more than 300 patients with refractory FM, CFS and CWP with a prescribed gluten-free diet showed remarkable clinical improvements in a large proportion of them [38].

Low-sugar low-yeast diets

A 24-week RCT in 52 CFS patients evaluated the effectiveness of a low-sugar and low-yeast (LSLY) diet in comparison to a healthy eating control group. The results were inconclusive as a result of the poor compliance, according to a systematic review [24]. Although there were no statistically significant differences in fatigue and quality of life between groups, the authors concluded that the healthy eating guidance is a more practical regime for these patients, given their difficulty in following the LSLY diet [24].

Organic foods and macrobiotic diets

Organic foods and macrobiotic diets have also been identified in MCS patients [13,19,25,26]. These all are common practices, but not clinically or scientifically proven.

### 3.2. Micronutrient Deficiencies and Nutritional Supplements

Numerous reviews reported that one of the hypotheses about the pathophysiology mechanism of CSS is the imbalance of trace metals, vitamins and certain minerals in human body fluids and tissues [8,9,10,19,27]. The CFS population is usually malnourished. So, the use of multivitamin–mineral supplements has been proposed as an easy approach [10,16]. However, a recent meta-analysis based on 27 studies by Joustra and colleagues doubts the specific role of micronutrient deficiencies in the pathophysiology of FM and CFS [27]. There is a lack of scientific evidence for whether MCS is associated with micronutrient deficiencies [33]. However, the limited available scientific evidence about this issue in FM, CFS and MCS shows a significant percentage of patients consume nutritional supplements [13,27].

Micronutrients related to redox equilibrium

Because of the high oxidative stress, total oxidative status and high level of reactive oxygen species that accompany these patients [10,39], some trace elements with implications in the cellular redox equilibrium, like magnesium, selenium and zinc, may have a possible involvement in the development of CSS. However, there are contradictory reports that oppose this hypothesis [8].

Hypomagnesemia is related to neurological and psychological symptoms and has been identified in FM and MCS patients [9,19]. Anxiety, stress or physical inactivity can trigger hypomagnesemia. Low intracellular magnesium in FM patients has been negatively correlated with adverse symptoms, such as muscle weakness and paresthesia [9]. Magnesium inhibits certain nerve receptors, such as N-methyl-D-aspartate (NMDA), possibly causing FM pain. Reduced levels of magnesium are associated with low-grade chronic systemic inflammation and may raise substance P, a neurokinin receptor agonist, related to the appearance of pain in FM patients. Likewise, magnesium levels are linked to chronic sleep deprivation, a common phenomenon in FM patients [9].

Moreover, a systematic review by Reid and colleagues involving a Grading of Recommendations, Assessment, Development and Evaluations (GRADE) valuation of the quality of evidence for interventions in CFS patients did not find differences in magnesium levels compared with the control subjects [28]. In addition, another study showed that supplementation with magnesium offered promising outcomes, although these results have not been replicated [31].

Selenium deficiency has been related to skeletal muscle disorders, due to an amplified prostaglandin synthesis [10] and many times has been found to be lower in patients with FM than control groups [9]. Selenium is part of the glutathione peroxidase (GPX) enzymes that are involved in the reduction of toxic peroxides. The exposure of patients to selenium-antagonistic toxic metals from the diet, such as Cd, inorganic Hg, Ni or Pb, could provoke a high burden and an exacerbation of symptoms linked to muscle pain [9,10].

Zinc deficiency may contribute to the dysfunction of natural killer cells and cell-mediated immune function. So, an evidence-based approach by clinicians for CFS diagnosis and management suggested that zinc supplements balanced with copper may be therapeutically promising for patients [16].

Iodine deficiency is considered a worldwide health problem, but without robust evidence that it is correlated with FM [8]. Iron may have a role in the etiology of FM, as a cofactor in serotonin and dopamine production [9]. One study has shown a higher prevalence of FM in people with iron deficiency anemia and thalassemia minor. Therefore, the authors highlighted the benefit of iron supplementation in these patients [8].

Supplementation with high antioxidant power substances, such as vitamin A, C and E, or coenzyme Q10 (CoQ10), may have beneficial outcomes for FM and CFS patients [8,10,12,27,32,39], mainly because of their protection against cell damage by oxidative stress and by assisting mitochondrial function. Given the gaps in the current scientific literature, further investigations are needed before recommendations are made about antioxidant supplementation [8,10,12]. Levels of circulating vitamin A are lower in FM and CFS patients in comparison to their control counterparts [27]. Although both vitamins A and E are promising treatments for CFS, further investigation is needed [10,27]. A systematic review and meta-analysis concludes that RCTs testing these supplements do not reveal clinical improvements with vitamin A supplementation in FM and CFS patients [27].

An 8-week RCT in 80 CFS patients showed that the CoQ10 plus nicotinamide adenine dinucleotide hydride (NADH) combination group had decreased fatigue and maximum heart rate during exercise, but without improved sleep or reduced pain, compared with the placebo group. However, the authors recommended additional and larger RCTs to confirm the results [30]. In contrast, oral NADH supplementation is not promising in CFS patients [28]. A systematic review by Campagnolo and colleagues recommended long-term controlled studies to evaluate the effectiveness of CoQ10 supplementation, with or without nicotinamide adenine dinucleotide hydride (NADH), for reducing fatigue in CFS patients because of an increase in cellular ATP synthesis through mitochondrial oxidative phosphorylation [12]. Indeed, a recent systematic review by Mehrabani and colleagues showed that the administration of 300 mg/day CoQ10 might alleviate fatigue in FM patients [32].

Micronutrients related to inflammation and the nervous system

Neuroinflammation has been suggested as an important factor in the pathogenesis of FM [39]. Inflammatory cytokines may play a role in fatigue, pain and stress [8]. Some substances, such as certain natural flavonoids, like luteolin and quercetin, have been considered as a possible treatment of CSS [39].

On the other hand, vitamin B12 metabolism has been proposed because of its implication in neurobehavioral dysfunction in CSS. Low levels of vitamin B12 have been found in cerebrospinal fluid in CFS [16] and FM patients [9]. Thus, vitamin B12/folic acid could be beneficial in the long-term against fatigue and other symptoms in CFS patients [10], although a review claimed the response is better when highly concentrated methylcobalamin is injected, combined with a high dose of oral folic acid [30]. Indeed, vitamin B12 or folate deficiency is commonly found among MCS patients [18].

Nevertheless, a case–control study aiming to examine potential biological mechanisms underlying MCS in 223 women with MCS versus 194 controls did not find deficiencies in vitamin B12 tissue levels or folate levels, but reported case–control differences for vitamin B6 and homocysteine. Although some clinicians have previously described a trend of low serum vitamin B6 levels in six out of 10 MCS patients, this study did not support this conclusion [33].

The impact of vitamin D deficiency is widely addressed in FM and CFS, but with controversial results. For example, patients with osteomalacia due to vitamin D deficiency have been misdiagnosed as having FM [34]. Vitamin D deficiency has been associated with excess weight, seasonality, insufficient sun exposure, skin phototype or insufficient physical activity [29]. Similarly, the mechanism of vitamin D in CWP is unclear, but it is well documented that vitamin D receptors and the hydroxylation of 25-OH vitamin D are present in several regions of the central nervous system [29,35]. Likewise, vitamin D is related to numerous pathways of regulation and modulation in pain, including receptors, neurotransmitters, cytokines, inflammation factors, vasodilation [29,35] and sleep–wake cycle modulation [34]. In addition, it has been suggested that vitamin D has immunomodulatory properties in autoimmune diseases [34] and influences pain pathways related to FM pathogenesis [29].

Numerous studies have shown a link between vitamin D and sleep and pain mechanisms. The role of vitamin D as a sleep modulator has been demonstrated, although the physiological mechanisms are unclear. Low vitamin D serum levels have been shown to correlate to short sleep duration [34]. Similarly, alterations in sleep contributes to the intensification of pain sensitivity. In fact, studies in FM patients have identified changes in sleep patterns, implying poor sleep quality, followed by subjective pain and fatigue [9,34]. Of note, sleep disorders can trigger painful conditions such as hyperalgesia [34].

Hypovitaminosis D has been shown in previous clinical trials in FM patients [29,34] and often been found to co-exist with anxiety and depression [9]. There are several reports that probe an inverse relationship between serum vitamin D levels and pain in FM patients [29,34]. Therefore, it has been suggested that, although it is difficult to establish an appropriate dose of vitamin D, its supplementation, combined with sleep hygiene, may be beneficial in FM patients [34]. A number of studies have detected hypovitaminosis in FM women, even though they have similar bone density as healthy women, implying that an association between vitamin D and pain in FM does not exist [8]. Karras and colleagues concluded that a discrepancy in the results could depend on several factors, such as differences between vitamin D supplements used in each trial (ergocalciferol or cholecalciferol), the adequate dosing and duration of supplementation and the concomitant use of calcium supplements. Consequently, although vitamin D seems to be a potential therapeutic strategy in FM, future large-scale and appropriately designed studies are still needed [29].

A retrospective study in CFS showed a high percentage of 25-OH vitamin in moderate to severe suboptimal serum levels, and recommended eating foods enriched in this vitamin, sun exposure and supplementation [24]. By contrast, an interventional study advised that oral vitamin D supplements of do not alleviate fatigue or improve markers of oxidative stress and inflammation in CFS [24].

A recent meta-analysis based on six RCTs by Yong and colleagues explores the effect of 25-OH supplementation on the management of chronic widespread pain and concluded that this supplementation could reduce pain scores after raising serum vitamin D levels [35].

Several reviews and meta-analyses in FM and CFS patients show insufficient scientific evidence for the routine recommendation of nutritional supplements in daily clinical practice and highlight personalized approaches [9,10,12,24,27,28,29,30]. The authors highlighted that the vast majority of studies concerning FMS patients are heterogeneous, observational and have fatal methodological weaknesses in their design. Besides, the lack of adjustment analyses for potential confounders, the establishment of control group selection and publication bias determination are common. Finally, the analysis is usually in isolated studies, without considering the high consumption of supplements that each CSS patient performs [27].

Regarding the applicability of the recent guidelines, a multidisciplinary approach is the best treatment for these diseases [40,41,42,43]. Multimodal management may include both nonpharmacological strategies (i.e., aerobic exercise, psychotherapy/cognitive behavioral therapy, defined physical and meditative movement therapies, strengthening exercise, among others) as first-line therapy, and pharmacological therapy, if it is necessary. However, according to the level of evidence of nonpharmacological approaches, different strengths of recommendation exist [40,41,42,43].

A personalized treatment, considering the severity of the disease symptoms, quality of life, coexistence with other diseases, pharmacological treatment and changing clinical characteristics, may be the best way to tailor specific dietary interventions in these patients.

It is striking that, despite the high prevalence of gastrointestinal and other diet-related problems and the impact nutritional treatment may have on these patients, none of these practical guides include dietary guidelines for the treatment of CSS patients [40,41,42,43].

There are no standard procedure algorithms for selecting appropriate treatments. Hence, it could be beneficial to include a complete dietary history in anamnesis. Some of the items to consider should be food frequency, preferences, intolerances, avoidances and triggers of adverse symptoms, among others. Additionally, the development of a nutritional screening and the analytical determination that assesses the presence of micronutrient deficiencies should be essential to determine the nutritional status of the patients.

The clinicians’ and dietitians’ performances should focus on calculating energy requirements to ensure a correct energy balance and recommend the consumption of specific foods according to their tolerance. Likewise, they should provide information about the severe consequences of diet manipulations without professional supervision.

One of the limitations in evaluating the effect of food and nutrition is the frequent overlap of several disorders within the same individual and relating the effect that each of these pathologies has on the patient’s quality of life. Moreover, studies focused on the influence of nutrition in MSC patients are very scarce. Furthermore, frequently published studies have severe methodological limitations, including a lack of appropriate controls, small sample sizes, non-standardized methodology or not reporting details of dietary assessment methods at baseline and during the study. Of note, a weakness in our revision process was the manual curation and the solo use of PubMed database as a search engine.

## 4. Conclusions

Existing available data show the complexity of accurately determining the role of diet in FM, CFS and MSC and simultaneously highlighting the need to continue the investigations in well-designed RCTs. Current indications show that dietary modifications may vastly improve the patients’ quality of life at a low cost. We suggest personalized treatment, in accordance with the severity of the disease symptoms, quality of life, coexistence with other diseases, pharmacological treatment, changing clinical characteristics, nutritional status, energy requirements and food tolerances, among others, as the best way to tailor specific dietary interventions in these patients. These approaches will partially overcome the challenges of the scientific and clinical research of FM, CFS and MSC. Patients should also be advised on the serious consequences of following dietary practices without health professionals’ supervision.

## Figures and Tables

**Figure 1 jcm-09-04106-f001:**
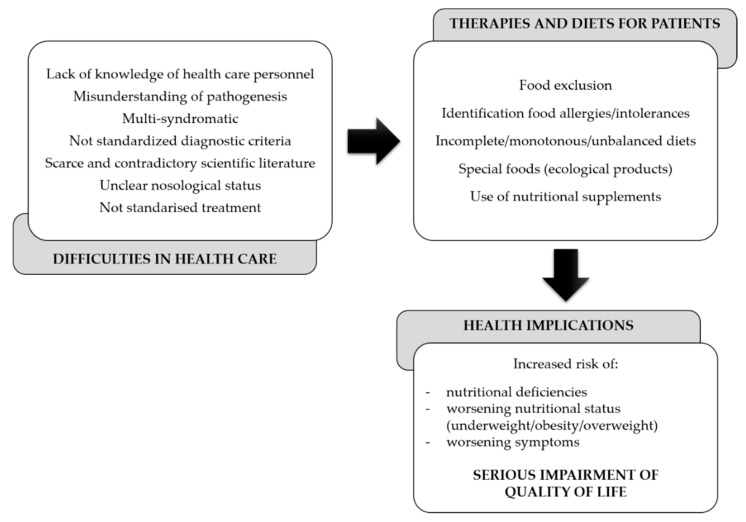
Gaps in the dietary and nutritional management of central sensitization syndromes (CSSs).

**Table 1 jcm-09-04106-t001:** Special dietary interventions and main conclusions according to the references.

Dietary Interventions	Syndrome	Main Conclusions	Author, Year [reference]
Hypocaloric diet	FM	Unclear effectiveness in normal-weight patients	Aman et al., 2018 [11]
	FM	Weight control seems to be a useful tool to improve the symptoms	Bested and Marshall, 2015 [16]
Elimination diet	FM	Inconclusive findings due to an elevated dropout rate of participants, a lack of random assignment, self-reported symptoms, poor adherence to the intervention	Li and Micheletti, 2011 [17]
	CFS	Insufficient evidence for use to relieve symptoms and risk of deficiencies in long-term use (>6 m)	Campagnolo et al., 2017 [12]
	MCS	Combination with avoidance of triggering environmental agents (chemical substances, sounds and lights, among others) is beneficial (a clinicians’ point of view)	Levin and Bayers, 1992 [18]; Ross, 1992 [19]
	MCS	A higher risk of deleterious effect (macronutrient imbalances and micronutrient deficiencies)	Aguilar-Aguilar et al., 2018 [13]
Avoidance of food additives	FM	Decreased consumption of monosodium glutamate (MSG) does not consistently reduce pain	Cairns, 2016 [20]
	FM	Regular daily consumption of aspartame (5 mg/kg) is not associated with pain conditions	Cairns, 2016 [20]
	MCS	Food additives containing azo dyes might play essential roles as elicitors in pediatric patients (a 5-year-old girl case report)	Inomata et al., 2006 [21]
Well-balanced diet	FM	A healthy diet is beneficial for FM symptoms	Bjørklund et al., 2018 [9]
	CFS	A well-balanced diet is essential for healing (clinicians’ point of view)	Bested and Marshall, 2015 [16]
	CFS	Eating a balanced diet and a variety of nutritious foods from the basic food groups according to the dietary guidelines for healthy people is the recommendation	Campagnolo et al., 2017 [12]
	CI	Maintaining a regular diet during 3 years of the follow-up period (n = 909) was significantly associated with improvement of pain	Azuma et al., 2019 [22]
Oligoantigenic	FM	Lack of scientific evidence	Arranz et al., 2010 [8]
Vegan and vegetarian diet	FM	Unclear effectiveness for pain treatment	Baranowsky et al., 2009 [23]
	FM	Likely beneficial effects due to the increase in antioxidant intake, but more complete studies are needed to confirm	Arranz et al., 2010 [8]
	FM	There is no sufficient evidence to support this dietary recommendation	Li and Micheletti, 2011 [17]
	FM	It is not clear whether symptom reduction is linked to the exclusion of many processed foods and the weight loss	Holton et al., 2009 [15]
Low-FODMAP diet	FM	Unclear effectiveness for pain treatment due to the high concurrence with IBS	Holton et al., 2009 [15]
	FM	Insufficient evidence for routine use. It might be beneficial in patients with significant distress from gastrointestinal symptoms	Aman et al., 2018 [11]
Gluten-free diet	FM	Insufficient evidence for routine use. It might be beneficial in patients with concurrent FM and gluten-related disorders	Aman et al., 2018 [11]
Low-sugar low-yeast diet	CFS	Inconclusive findings in symptom alleviation due to the lack of rigor of designed studies	Jones and Probst, 2017 [24]
Organic foods and macrobiotic diet	MCS	Lack of scientific evidence of its effectiveness or safety, and clinically unproven in the improvement of symptomatology	Nogué Xarau et al., 2011 [25]; McGraw, 2011 [26] Aguilar-Aguilar et al., 2018 [13]

CFS: chronic fatigue syndrome; CI: chemical intolerance; FM: fibromyalgia; FODMAP: fermentable, oligo-, di- and monosaccharides and polyols; IBS: irritable bowel syndrome; MCS: multiple chemical sensitivity.

**Table 2 jcm-09-04106-t002:** Micronutrient deficiencies, nutritional supplements and main conclusions according to the references.

Micronutrient Deficiencies and Nutritional Supplements	Syndrome	Main Conclusions	Author, Year [reference]
**General**			
Micronutrient imbalances	FM/CFS/MCS	The current evidence links CSS to several micronutrient imbalances, but the implications in the development and pathophysiology are unclear	Ross, 1992 [19]; Arranz et al., 2010 [8]; Joustra et al., 2017 [27]; Bjørklund et al., 2018 [9]; Bjørklund et al., 2019 [10]
Multivitamin–mineral supplementation	FM/CFS	Likely a safe and straightforward approach to alleviate the symptoms and improve quality of life, but nowadays, findings for routine recommendation are inconclusive	Reid et al., 2011 [28]; Karras et al., 2016 [29]; Campagnolo et al., 2017 [12]; Jones and Probst, 2017 [24]; Joustra et al., 2017 [27]; Castro-Marrero et al., 2017 [30]; Bjørklund et al., 2018 [9]; Bjørklund et al., 2019 [10]
**Related to redox equilibrium**			
Magnesium deficiency	FM	Reduced intracellular Mg content is frequent. It has a deleterious effect on symptoms	Bjørklund et al., 2018 [9]
	CFS	Non-existent differences in plasma and blood Mg compared with controls, although there are discrepancies in the establishment of normal ranges	Reid et al., 2011 [28]
	MCS	Mg deficiency is one of the most prevalent micronutrient deficiencies	Ross, 1992 [19]
Magnesium supplementation	CFS	Promising effectiveness in single studies, but the evidence remains very limited	Chambers et al., 2006 [31]
Selenium deficiency	FM	Se status is significantly reduced compared to healthy people. It has a deleterious effect on symptoms (fatigue, muscle pain and weakness)	Bjørklund et al., 2018 [9]
Zinc + copper supplementation	CFS	Promising effectiveness on adverse symptomatology	Bested and Marshall, 2015 [16]
Iodine deficiency	FM	Non-existent total evidence	Arranz et al., 2010 [8]
Iron deficiency and supplementation	FM	Anemia and thalassemia minor are common. Promising beneficial supplementation in patients with concurrence of these diseases related to iron	Arranz et al., 2010 [8]
Levels of vitamin A and E, and supplementation	FM/CFS	Reduced levels are recurrent. Little evidence of beneficial supplementation. Potential treatment that needs further examination	Joustra et al., 2017 [27]; Bjørklund et al., 2019 [10]
CoQ10 supplementation	FM	Beneficial (300 mg/day) in alleviating fatigue	Mehrabani et al., 2019 [32]
	CFS	Promising effectiveness in reducing fatigue (with or without NADH)	Castro-Marrero et al., 2017 [30]; Campagnolo et al., 2017 [12]; Mehrabani et al., 2019 [32]
NADH supplementation	CFS	There is no good evidence for alleviating fatigue	Reid et al., 2011 [28]
**Related to inflammation and nervous system**			
Levels of vitamin B12	FM	Reduced levels are common	Bjørklund et al., 2018 [9]
	CFS	Low levels in cerebrospinal fluid are frequent	Bested and Marshall, 2015 [16]
	MCS	Frequent deficiency according to clinicians’ point of view	Levin and Byers, 1992 [18]
	MCS	Non-existent low tissue levels (case–control study)	Baines et al., 2004 [33]
Levels of folates	MCS	Frequent deficiency according to clinicians’ point of view	Levin and Byers, 1992 [18]
	MCS	Non-existent low tissue levels (case–control study)	Baines et al., 2004 [33]
Vitamin B12/folic acid	CFS	A dose–response association and long-lasting effects provide a proper positive reaction	Bjørklund et al., 2019 [10]
	CFS	Promising effectiveness when highly concentrated methylcobalamin is injected combined with a high dose of oral folic acid	Castro-Marrero et al., 2017 [30]
Levels of vitamin B6	MCS	Serum levels are within normal ranges, but still lower than controls (case–control study)	Baines et al., 2004 [33]
Levels of vitamin D	FM	Hypovitaminosis D is common	Arranz et al., 2010 [8]; Karras et al., 2016 [29]; de Oliveira et al., 2017 [34]
	CFS	Suboptimal serum levels are habitual	Jones and Probst, 2017 [24]
Vitamin D supplementation	FM	Undefined dose supplementation and combination with sleep hygiene seem to be a potential therapeutic strategy	de Oliveira et al., 2017 [34]
	FM	Potential beneficial effects on pain and severity, but future large-scale studies are needed to determine the optimal dosing and duration	Karras et al., 2016 [29]
	FM	Promising effectiveness in ameliorating pain, but inconclusive findings due to divergence of results	Arranz et al., 2010 [8]
	CFS	Promising effectiveness in improvement of fatigue and selected markers of inflammatory and oxidative stress	Jones and Probst, 2017 [24]
	CWP	Promising effectiveness in reducing pain	Yong et al., 2017 [35]

CSS: central sensitization syndrome; CFS: chronic fatigue syndrome; CWP: chronic widespread pain; FM: fibromyalgia; MCS: multiple chemical sensitivity.

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
