# Peer review of "Food Implications in Central Sensitization Syndromes"

_jcm, 2020, doi:10.3390/jcm9124106_

Round 1
Reviewer 1 Report
I congratulate the authors for conducting a systematic review of a subject that has become important in central sensitization syndromes management. This review is useful for clinicians to show what is already known, and it is also a guide for future research.
Even clarification of syndromes was nicely given in the introduction, the importance of dietary components usage for these syndromes is not explained well. A possible relationship between dietary components and these syndromes could be clarified and highlighted more clear to strength the rationales behind this study. For instance; "The use of dietary components with potential functionality on inflammation and oxidative stress deem promising in the treatment of these patients. (lines 70-72)" part could be supported by some examples or explanations.
In lines 355-356, the word of "nuts" is repeated (..., nuts, shellfish, solanaceous plant species, nuts, soy, yeast and arachidonic acid [12,14,15,27].), please delete one of them.
Author Response
Dear reviewer,
We would like to thank you for your constructive comments and suggestions in our manuscript entitled “Food implications in central sensitization syndromes” (Ref: Manuscript ID jcm-1014608) by Aguilar-Aguilar and colleagues. Please see the attachement.
We hope that you find our revised work suitable for publication in JCM.
We look forward to receiving your final decision.
Yours sincerely,
Dr. Elena Aguilar-Aguilar

Reviewer 2 Report
This is a well written review article, about the role of food in CSS. Data are well presented and analyzed.
Table 1 is quite interesting as it sums up authors’ findings on dietary components in CSSs.
Figure A (Appendix) clarifies the methodology and tools used for data collection.
The conclusion and main findings are not novel, but the paper presents thought-provoking approaches for further study.
Author Response
Dear reviewer,
We would like to thank you for your constructive comments and suggestions in our manuscript entitled “Food implications in central sensitization syndromes” (Ref: Manuscript ID jcm-1014608) by Aguilar-Aguilar and colleagues.
We hope that you find our revised work suitable for publication in JCM.
We look forward to receiving your final decision.
Yours sincerely,
Dr. Elena Aguilar-Aguilar
Reviewer 3 Report
Food implications in central sensitization syndromes
The objective of this manuscript was to gather information about the current status of dietary recommendations and discuss the scientific evidence in order to shed light on nutritional management for the CSS patients.
The objective under consideration is interesting, since it reviews data on dietary modifications in relation to CSS. However, 16 of the 41 references included, are reviews regarding the role of specific ingredients/foods or the whole diet on CSS. Consequently, several questions are raised regarding the novelty of the manuscript.
Moreover, I strongly believe that the authors must revise and improve the entire manuscript taking into consideration all the comments below.
General comments
- I would greatly appreciate if, the authors could inform us whether the purpose of the manuscript is indeed a. to present the contradictory results found in the literature concerning the effect of dietary modifications on CSS and b. to confirm the necessity of personalized treatment under the supervision of a nutritionist or clinician. If this is the case, the objective as well as the whole manuscript has to be rearranged in a more clear way.
- In my opinion, the scientific literature is not scarce, as it appears on line 14; but there is a number of studies available as described also in reference [24].
- Are there any specific dietary guidelines for patients suffering from CSS?
- Introduction: Although the introduction section is quite well-written, some parts are not well-structured and should be rearranged. Repetitions are also found in the following sentences; for example in lines 56-57 and 63-64 as well as in lines 59-60 and 65-68, respectively.
- Methods: I strongly believe that the authors should provide more information on the methodology followed. In this frame, I would greatly appreciate if the authors give more data regarding the inclusion criteria and particularly the populations under study. According to Jones & Evans (2000): “The population of participants to be included for review should be predetermined and will make explicit the types of people, their disease or health condition and the setting of interest for review [Jones, T., & Evans, D. (2000). Australian Critical Care, 13(2), 66-71].
- Furthermore, it should be emphasized that the critical evaluation of the individual studies selected is a fundamental step in the review process. Consequently, the fact that the authors did not perform any quality assessment, especially in the case of interventional studies, is a significant flaw of the study.
- Sections 3.1.1-3.1.4: In my opinion, this part is rather wordy and in some parts difficult to follow. Please rewrite in a more accurate way highlighting only the most significant data.
- I fully agree with the authors, that the existing available data show the complication to accurately determine the role of diet in FM, CFS and MSC since the findings presented in lines 147-148, 175-176, 179-180, 187-180, 210-211, 225-227, 261-262, 284-285 are quite inconclusive. Therefore, I strongly suggest the authors to shed light on nutritional management for the CSS patients by implementing a “what-if scenario” perhaps under the guidance of a nutritionist or a clinician.
Specific comments
- Line 14: “scientific literature is scarce”. Based on the given literature, please rephrase for clarity.
- Line 20: “personalized treatment”. Please give more details in the whole manuscript.
- Lines 23-26: Please rewrite in a more clear way.
- Line 48: “contributes to adverse reactions”. Please consider rephrasing for clarity.
- Line 53: “genetic profiles of impairment"”. Please explain in a more accurate way.
- Line 58: Please consider specifying the term “mucosa” in order to be more precise.
- Lines 65-66: Please rewrite in order to be more accurate.
- Lines 80-82: Please consider rephrasing for clarity.
- Lines 103-104: “We manually revised and included Spanish and English written papers as well as clinical human studies”. Please rewrite to be more accurate.
- Lines 114-116: Please explain in a more clear way.
- Lines 133-140: Please move this part to a new paragraph.
- Lines 135-140: Please rephrase for clarity.
- Line 150: Please explain the term “Chemical intolerance” based on the given literature.
- Lines 154-155: Please explain in more details.
- Lines 163-164: Please rephrase for clarity.
- Lines 166-167: As lutein and lycopene, are not precursors of vitamin A in mammals, please rephrase.
- Lines 175-176: Please move this sentence to the end of the next paragraph.
- Lines 179-181: “loss of weight in patients may have influenced the outcomes”. In order to be highly informative, please comment on this observation.
- Lines 182-185: More information regarding the arms of the study should be provided. Furthermore, please provide references in line 185.
- Lines 182-188: The effects of vegetarian diets as well as the conclusions regarding the adoption of these diets are discussed also in lines 169-176. Please rearrange the structure.
- Lines 194-202: Please rewrite in a more clear way. Furthermore, please provide references in line 196.
- Lines 203-209: Please check ref. [24] and if necessary rephrase. Furthermore, if necessary please provide references regarding the 24-week RCT.
- Lines 215-217: Please explain in more details.
- Lines 238-239: Please rephrase for clarity.
- Line 293: “So, the authors recommend a larger RCT”. Please explain in more details.
- Lines 295-308: The text structure is quite confusing for the readership and difficult to follow. Please rewrite.
- Line 305: “other nutrients”. In my opinion, all nutrients should be discussed in the same section.
- Lines 354-356: In my opinion, the term “ingredients” is not the appropriate based on the examples given.
- Line 364: Please omit “intake”.
- Line 390: Please check ref. [24] and if necessary rephrase.
- Line 407: “a typology with rigorousness”. Please explain in more details.
- Lines 410-411: Please explain in more details. Furthermore, significant questions are raised regarding the authors’ choice to use only the PubMed database.
- Lines 419-421: Please rewrite in a more clear way.
Table 1:
- The table is quite extensive and does not facilitate the readership. I strongly believe that the authors should split the initial table into 3 distinctive tables for diet, nutrient intake, and supplements. Furthermore, review articles should be omitted.
- Population characteristics as well as a short description of each diet should be provided. In case of interventional studies, the arms of the study as well as the administered dose should be provided.
- Please rewrite the main conclusions in more detail to be highly informative.
- Please define the exact duration of the long-term period.
Author Response
Dear reviewer,
We would like to thank you for your constructive comments and suggestions in our manuscript entitled “Food implications in central sensitization syndromes” (Ref: Manuscript ID jcm-1014608) by Aguilar-Aguilar and colleagues. Please see the attachment.
We hope that you find our revised work suitable for publication in JCM.
We look forward to receiving your final decision.
Yours sincerely,
Dr. Elena Aguilar-Aguilar

Round 2
Reviewer 3 Report
Food implications in central sensitization syndromes
This manuscript aims to gather information about dietary recommendations’ current status and discuss the scientific evidence in detail to shed light on nutritional management for CSS patients.
The authors have taken into consideration the previous comments and the appropriate changes have been made. Nevertheless, this revised version is very difficult to read due to the track-changes and the struck-out words included in the pdf file.
The presentation of the sections entitled “Methods” and “Dietary components in these syndromes” have been significantly improved. Meanwhile, the tables’ format as well as the data being provided in the tables, have been modified in order to facilitate the readership.
However, before publication, the authors should consider the comments below in order to further strengthen the presentation of the manuscript.
-
Line 14: “multi-syndromatic patients” Please check this term and if necessary correct this term in the whole manuscript.
-
Lines 23-28: Please consider omitting “micronutrients deficiencies”, since the term “nutritional status” refers to the state of a person’s health in terms of the nutrients in his/her diet.
-
Lines 65-67: “52 MCS participants (all women) and 52 healthy volunteers” Please rephrase for clarity. Furthermore, please check data provided in ref [6]
-
Line 67-70: Please clarify whether ref [6] corresponds to Micarelli et al or Loria-Kohen et al.
-
Lines 139-140: “written papers as well as clinical human studies”. Although the authors have taken into consideration the suggestions provided, the difference between the terms “written papers” and “clinical human studies” is still to be defined. Please rephrase.
-
Line 167: “patients' techniques” Please rephrase for clarity
-
Lines 212-222: Please change the references format according to the journal guidelines. Furthermore, these references should also be cited at the references list at the end of the manuscript
-
Lines 269-275: Please rephrase for clarity
-
Lines 280-282: Please rewrite to be more accurate
-
Lines 296-304: Please rewrite in a more clear way and provide the appropriate references
-
Line 317: Please change the reference format according to the journal guidelines. Furthermore, these references should also be cited at the references list at the end of the manuscript.
-
Lines 340-342: Please rephrase for clarity.
-
Line 510: “are not heterogeneous”. Please check and if necessary correct.
-
Line 610: Please change the reference format according to the journal guidelines. Furthermore, these references should also be cited at the references list at the end of the manuscript.
-
Lines 611-616: Please rewrite in a more clear way.
-
Line 632: “severe methodological”. The phrase should change as: severe methodological limitations.
Table 1:
-
“environmental control” Please explain in more detail
-
“Unclear responsible for effectiveness” Please rewrite in a more clear way
-
“Making specific dietary recommendations for treatment is not possible with limited available data” Please rephrase for clarity.
Table 2:
-
“Non-existing low serum levels” Please rephrase to be more accurate
Author Response
Dear reviewer:
We want to thank your constructive comments and suggestions in our manuscript entitled “Food implications in central sensitization syndromes” (Ref: Manuscript ID jcm-1014608) by Aguilar-Aguilar and colleagues.
Please see the point-by-point response.
- Line 14: “multi-syndromatic patients” Please check this term and if necessary correct this term in the whole manuscript.
We have changed the term “multi-syndromatic” for “multi-syndromic” [Lines: 14, 71-72, 74].
- Lines 23-28: Please consider omitting “micronutrients deficiencies”, since the term “nutritional status” refers to the state of a person’s health in terms of the nutrients in his/her diet.
Since the term "nutritional status" includes the evaluation of possible nutritional deficiencies, we have omitted "nutritional deficiencies" in the abstract [Line: 24] and in the conclusions sections [Line: 505].
- Lines 65-67: “52 MCS participants (all women) and 52 healthy volunteers” Please rephrase for clarity. Furthermore, please check data provided in ref [6]
According the reviewer’s suggestion, we have rewritten this sentence as follows, and we have checked the reference.
“A case in point, statistically significant differences in the frequencies of MTHFR rs1801133 were observed in a descriptive study composed of 52 MCS patients and 52 healthy volunteers by Loria-Kohen et al” [Lines: 59-61].
“6. Loria-Kohen, V.; Marcos-Pasero, H.; de la Iglesia, R.; Aguilar-Aguilar, E.; Espinosa-Salinas, I.; Herranz, J.; Ramírez de Molina, A.; Reglero, G. Multiple chemical sensitivity: Genotypic characterization, nutritional status and quality of life in 52 patients. Med Clin (Barc) 2017, 149, 141–146, doi:10.1016/j.medcli.2017.01.022” [Lines: 544-547].
- Line 67-70: Please clarify whether ref [6] corresponds to Micarelli et al or Loria-Kohen et al.
We apologize for the confusion with the two references. We had a performance problem between the bibliographic manager Zotero and the Word change control in the revised version and the modifications to the references were only included in the final manuscript. However, we have verified that they are correctly included in this document, and we have highlighted the additions in red in the references’ section.
“2. Micarelli, A.; Cormano, A.; Caccamo, D.; Alessandrini, M. Olfactory-Related Quality of Life in Multiple Chemical Sensitivity: A Genetic-Acquired Factors Model. Int J Mol Sci 2020, 21, 156, doi:10.3390/ijms21010156” [Lines: 530-532].
“6. Loria-Kohen, V.; Marcos-Pasero, H.; de la Iglesia, R.; Aguilar-Aguilar, E.; Espinosa-Salinas, I.; Herranz, J.; Ramírez de Molina, A.; Reglero, G. Multiple chemical sensitivity: Genotypic characterization, nutritional status and quality of life in 52 patients. Med Clin (Barc) 2017, 149, 141–146, doi:10.1016/j.medcli.2017.01.022” [Lines: 544-547].
- Lines 139-140: “written papers as well as clinical human studies”. Although the authors have taken into consideration the suggestions provided, the difference between the terms “written papers” and “clinical human studies” is still to be defined. Please rephrase.
We have rewritten this sentence and relocated the term “clinical” at the beginning of this section.
“Clinical human articles associated with central sensitization disorders, (…)” [Line: 103].
“We manually revised and included Spanish and English written papers” [Line: 126].
- Line 167: “patients' techniques” Please rephrase for clarity
We have changed the word “techniques” for “dietary interventions” as follows.
“(…) are some of the most used CSS patients' dietary interventions” [Line: 156].
- Lines 212-222: Please change the references format according to the journal guidelines. Furthermore, these references should also be cited at the references list at the end of the manuscript
As abovementioned, we have verified the insertion of new references, and revised the bibliography section.
- Lines 269-275: Please rephrase for clarity
We have rewritten the text.
“The Living Foods diet (LFD) is a raw vegan diet that contains a high quantity of antioxidants [8], including berries, cereal, fruits, germinated seeds, nuts, roots, sprouts, and vegetables. In addition, it is free of alcohol, coffee table salt, and tea [15]. The LFD may improve joint stiffness and pain in FM patients due to higher levels of polyphenolic compounds, (i.e., kaempherol, myricetin, and quercetin, and vitamins such as A (α and β-carotenes), C and E) than their counterpart controls [8]” [Lines: 252-258].
- Lines 280-282: Please rewrite to be more accurate
According to the reviewer’s comment, we have rewritten the text to be more accurate.
“In a 3-month’s non-randomized, controlled study, the LFD diet showed beneficial effects on symptoms, such as morning stiffness, pain scores, and sleep quality as well as in health assessment questionnaire scores in FM patients. However, it was also recorded that FM subjects had lost weight” [Lines: 263-265].
- Lines 296-304: Please rewrite in a more clear way and provide the appropriate references
We have rewritten this paragraph and provided the references.
“Some studies have compared the effect of vegan or vegetarian diets with pharmacological treatments [8,17,23]. Regarding some reviews, a 6-week RCT with two arms evaluated the effects of a vegetarian diet exclusively in comparison to the administration of a tricyclic antidepressant (10-25 mg/day, tittered up to 100 mg/day during the study) in 78 FM patients. A decreased in pain scores was observed in both groups, but with a lower trend for the diet’s group [8,17,23]” [Lines: 275-279].
- Line 317: Please change the reference format according to the journal guidelines. Furthermore, these references should also be cited at the references list at the end of the manuscript.
We have verified the insertion of new references, and completely revised the bibliography section.
- Lines 340-342: Please rephrase for clarity.
We have rephrased this paragraph.
“A 24-week RCT study in 52 CFS patients evaluated the effectiveness of a low sugar and low yeast diet (LSLY) in comparison to a healthy eating control group. The results were inconclusive as a result of the poor compliance, according to a systematic review [24]” [Lines: 302-304].
- Line 510: “are not heterogeneous”. Please check and if necessary correct.
We have ammended the sentence as follows.
“The authors highlighted that the vast majority of studies concerning FMS patients, are heterogeneous, observational (…)” [Lines: 450-451].
- Line 610: Please change the reference format according to the journal guidelines. Furthermore, these references should also be cited at the references list at the end of the manuscript.
As abovementioned, we have verified the insertion of new references, and revised the reference’s section.
- Lines 611-616: Please rewrite in a more clear way.
Following the reviewer's recommendation, we have rewritten the paragraph to be more accurate.
“There are no standard procedure algorithms for selecting appropriate treatments. Hence, it could be beneficial to include a complete dietary history in anamnesis. Some of the items to consider should be food frequency, preferences, intolerances, avoidances, and triggers of adverse symptoms, among others. Additionally, the development of a nutritional screening and the analytical determination that assesses the presence of micronutrient deficiencies should be essential to determine the nutritional status of the patients” [Lines: 469-474].
- Line 632: “severe methodological”. The phrase should change as: severe methodological limitations.
We have introduced the suggested word.
“Furthermore, frequently published studies have severe methodological limitations, (…)” [Line: 494].
Table 1:
- “environmental control” Please explain in more detail
We have included additional information about this concept as follows.
“Combination with avoidance of triggers environmental agents (chemical substances, sounds, and lights, among others) are beneficial (a clinicians’ point of view)”.
- “Making specific dietary recommendations for treatment is not possible with limited available data” Please rephrase for clarity.
We have rephrased to clarify.
“There is no sufficient evidence to support this dietary recommendation”.
- “Unclear responsible for effectiveness” Please rewrite in a more clear way
We have rewritten this sentence.
“It is not clear whether symptom reduction is linked to the exclusion of many processed foods and the weight loss”.
Table 2:
- “Non-existing low serum levels” Please rephrase to be more accurate
We have explained this sentence.
“Serum levels are within normal ranges, but still lower than controls”.
Yours sincerely,
Dr Elena Aguilar-Aguilar
